# Odontogenic Sinusitis with Oroantral Communication and Fistula Management: Role of Regenerative Surgery

**DOI:** 10.3390/medicina59050937

**Published:** 2023-05-12

**Authors:** Lorenzo Sabatino, Michele Antonio Lopez, Simone Di Giovanni, Michelangelo Pierri, Francesco Iafrati, Luigi De Benedetto, Antonio Moffa, Manuele Casale

**Affiliations:** 1Unit of Integrated Therapies in Otolaryngology, Fondazione Policlinico Universitario Campus Bio-Medico, Via Alvaro del Portillo, 00128 Rome, Italy; l.debenedetto@policlinicocampus.it (M.A.L.);; 2Specialised Odontostomatology Department of Head and Neck and Sensory Organs, Division of Oral Surgery and Implantology, Fondazione Policlinico Universitario A. Gemelli IRCCS-Università Cattolica del Sacro Cuore, 00168 Rome, Italy; 3Unit of Integrated Therapies in Otolaryngology, Department of Medicine and Surgery, Università Campus Bio-Medico di Roma, Via Alvaro del Portillo, 00128 Rome, Italy; simone.digiovanni@unicampus.it (S.D.G.);

**Keywords:** odontogenic sinusitis, oro-antral fistula, oro-antral communication

## Abstract

*Objective*: The aim of this study is to show our experience with the correct management of patients suffering from odontogenic sinusitis with oroantral communication and fistula. *Methods*: According to the inclusion criteria, 41 patients were enrolled in this retrospective study with a diagnosis of odontogenic sinusitis with oroantral communication and fistula; 1 patient with pre-implantological complication, 14 with implantological complications, and 26 with classical complications. *Results*: Two patients were treated with a fractioned combined approach, 13 patients were treated with an oral approach only, and 26 patients were treated with a combination. There was a complete resolution of the symptoms and closure of the fistula in all the patients enrolled. *Conclusions*: In our study, in all 41 patients, there was a surgical success. The best option is to use a multidisciplinary approach for patients suffering from odontogenic sinusitis.

## 1. Introduction

Maxillary sinusitis is defined as symptomatic inflammation of the maxillary sinus. When the inflammation exceeds 12 weeks, it is classified as chronic [1]. Sinusitis is predominantly of rhinogenic origin, but dental infection can be a major predisposing factor in some cases [2].

Odontogenic sinusitis (OS) is characterized by the presence of a sinusal disease in which radiographic, microbiological, and/or clinical evidence indicates the dental origin of the disease [3]. In the last decade, we assisted an increasing OS prevalence determined by spreading in oral implantological and pre-implantological surgery [4]. Moreover, a better comprehension of the disease led to a greater diagnostic rate and reduced undetected cases.

When odontogenic sinusitis is misdiagnosed, inappropriate treatments may be used, which could lead to the progression of pulpal or implantological disease and the development of pansinusitis, osteomyelitis, meningitis, or, in extremely rare circumstances, involvement of the periorbital region with blindness [5,6].

According to the literature, odontogenic sinusitis accounts for 10–12% of all sinusitis cases, but recent studies suggest that it could be as high as 41% [7,8]. There are no standard protocols concerning its diagnosis and management. To our knowledge, Felisati et al. conducted the only study that has tried to systematize the approach to OS, proposing a new classification system based on OS etiology [3].

Apical and marginal periodontitis, oroantral communications or fistulas after tooth extraction, and infection caused by intra-antral foreign bodies are the most common causes of odontogenic sinusitis (OS) [7].

An oroantral communication (OAC) is defined as an open connection between the oral cavity and the maxillary sinus. When this communication is not adequately treated, it can develop into an oro-antral fistula (OAF), with the failure of spontaneous closure and the pathological epithelization of the communication [4]. OACs are mostly caused by the extraction of maxillary posterior teeth, especially when a peri-apical abnormality is present. Other common causes of OACs are complications from implant surgery, trauma, cysts, and tumor asportation [9].

The nasal cavity and oral cavity have, respectively, their own microbiome, so an OAC/OAF can be the cause of odontogenic sinusitis, causing the passage of bacteria from the oral cavity to the maxillary sinus [10]. Thus, despite the mainstay of treatment for odontogenic sinusitis being surgical therapy, antibiotics play an important role whenever they are combined with other appropriate treatments. Patients with odontogenic sinusitis have a larger and more diverse microbiological burden than those with CRS alone, and antimicrobial therapy should address this difference. Saibene et al. showed that 70% of odontogenic sinusitis isolates were susceptible to amoxicillin clavulanate [10]. For individuals with a penicillin allergy who cannot receive amoxicillin, doxycycline is the most appropriate treatment [11,12].

In most cases, OACs may close spontaneously, especially when the defect has a size smaller than 5 mm [13]; when this does not happen, closure of OACs is advised [14]. The closure is usually performed by surgical procedure, according to the size of the OAC/OAF, and can be achieved using different methods. According to Visscher et al., several techniques are proposed in the literature, and the most common one is the buccal advancement flap procedure [9].

As we know from the literature, odontogenic sinusitis is a heterogenous pathology due to the different clinical presentations, different signs or symptoms, and the anatomical zone affected. The clinical approach to OS is often challenging because of the heterogeneous clinical presentation. In addition, patients’ signs and symptoms are usually mild and atypical, leading to a misdiagnosis if the pathology is not directly sought [15].

The aim of this manuscript is to evaluate the outcome of the conservative surgical treatment for OAC.

## 2. Materials and Methods

A retrospective analysis was performed on the clinical data from the patients surgically treated for odontogenic sinusitis with OAC in the Department of Integrated Therapies in Otolaryngology (Campus Bio-Medico, Rome, Italy) from December 2020 to June 2021. Clinical data on etiology, therapy, and demographics were gathered.

The patients were enrolled considering the following inclusion criteria: (1) clinical diagnosis of sinusitis with suspected odontogenic etiology, supported by radiological and/or endoscopic findings and with medical treatment resistance; (2) presence of an oroantral fistula/communication (3) surgical treatment via oral approach with or without nasal approach (functional endoscopic sinus surgery, FESS); and (4) presence of computer tomography (CT) executed before the surgery. 

According to the inclusion criteria, we enrolled 41 consecutive patients diagnosed with odontogenic sinusitis with oroantral communication or Fistula. 

All patients underwent conservative medical therapy with at least 2 antibiotics courses, of which one was amoxicillin + clavulanic acid 1 g bid for 7 days and nasal irrigations. All patients with spontaneous oro-antral fistula closure after 7–10 days were excluded. 

A dental scan and a maxillofacial CT (Computer Tomography) scan were indicated for all patients to diagnose the involvement of the maxillary sinus and to study the anatomy of the nose and paranasal sinuses in order to program the correct treatment protocol. 

The diagnosis of sinusitis was made according to epos 2020 guidelines, and thus, signs and symptoms typically associated with this pathology (purulent rhinorrhoea, anterior and/or posterior, unilateral or bilateral nasal obstruction, and maxillary pain), confirmed by endoscopy and/or CT scan, appeared with the presence of OAC and did not respond to medical treatment, usually consisting topical nasal decongestants or steroids, mucolytic therapy, and systemic antibiotics [16]. Sinonasal neoplasms or other sinonasal entities, such as Schneiderian papilloma, were excluded as they require more extensive treatment with different post-operative impacts on quality of life [17,18].

Additionally, Felisati et al.’s classification [3] was used to divide the patients into 3 groups: pre-implantological complications, implantological complications, and classic complications.

The patients were divided into groups between acute or chronic OS, based on time criteria (3 months) and on the type of surgery used as treatment. 

Oral approach with removal of epithelial tissue in the communication, scarring of the margins, and first intention closure with local mucosal trapezoidal flap to mucosally close the fistula. In some cases, depending on the size of the defect, we also used a collagen sponge to aid the mucosal flap closure.


**
Surgical procedures
**



**FESS/mini-Caldwell–Luc**


For the surgical resolution of odontogenic sinusitis, the nasal approach consisted of a minimal functional endoscopic sinus surgery (FESS) performed under general anesthesia (GA), with the aim of addressing any significant anatomical variation causing a reduction in the sinus drainage (osteomeatal complex obstruction, severe deviated septum, polyps, and concha bullosa), opening the maxillary sinus with an uncinectomy and a middle meatal antrostomy, and cleaning of the maxillary sinus of pathological mucus or tissue. The natural ostium is surgically widened (Figure 1), and only infected sinus mucosa is removed, leaving the basement membrane alone. As a result, natural sinus mucosa is preserved, and mucociliary clearance is unaffected. This procedure necessitates a high level of experience and precision for the close proximity to anatomical structures such as the orbital nerve and eyes [19].

In selected cases, a mini-Caldwell–Luc approach is also used when better access to the sinus is required, for example, for removing large foreign bodies.


**Techniques of OAC/OAF closure**


All surgeries were performed by the same operator, who has experience in the treatment of oro-antral communications, ensuring the standardization of the technique.

Before each surgery, patients orally took Amoxicillin 875 mg/Clavulanic Acid 125 mg on the morning of surgery and continued twice daily, for a total of five days. The surgeon administered a local anesthetic (Articaine hydrochloride 4% with adrenaline 1:100,000) before operating.

With regard to the anesthetic technique used, local anesthesia was performed starting from the most distal area. First, the posterior superior alveolar nerve was vestibularly anesthetized. Then, more mesially, in the area of the fourth, the infraorbital nerve. Moving palatally, anesthesia of the greater palatine nerve and the nasopalatine nerve was carried out. When necessary, anesthesia was strengthened in areas still sensitive to the pain sensation. This sequence of anesthesia allows us to obtain excellent results by making the last ones less annoying at the palatal level.

### 2.1. Technique with Local Flaps

Local buccal soft tissue flaps are often indicated in the closure of small to moderate-sized defects. We used this technique for defects between 5–8 mm. A large trapezoid mucoperiosteal flap is elevated and is used to primarily close the defect. Non-absorbable 4/0 threads in pseudomonofilament of polyamide were utilized for the sutures to achieve closure by primary intention.

### 2.2. Technique with Cortico-Cancellous Graft Covered with Resorbable Collagen Membranes

This technique was used for the closure of defects bigger than 8 mm with sufficient adjacent bone vestibularly and palatally to the defect. A full-thickness mucoperiosteal flap was raised using a periosteal elevator after a crestal incision was made using a scalpel and a 15c blade. In this way, it was possible to increase the visibility of communication and facilitate access to the sinus. Using a syringe without a needle to draw blood from the flap incision and setting it aside to mix with the heterologous cortico-cancellous bone may be helpful during this stage. A heterologous cortico-cancellous graft was implanted vestibularly and palatally inside the maxillary sinus after the Schneider membrane was detached (GenOs Osteobiol^®^, Turin, Italy) and coated with a resorbable collagen membrane (Evolution Osteobiol^®^, Turin, Italy) (Figure 2). By using the flap or, if necessary, pins, the membrane was positioned excessively and stabilized on the crestal walls of the defect. The palatine and vestibular walls of the maxillary sinus were placed in conjunction with the graft with great care. A continuous para-crestal suture and two detachable stitches were used to close the incision. Non-absorbable 4/0 threads in pseudomonofilament of polyamide were utilized for the sutures to achieve closure by primary intention.

### 2.3. Technique with Cortico-Cancellous Graft Covered with Resorbable Collagen Membranes and Heterologous Cortical Lamina

This technique was used for the closure of defects bigger than 8 mm that required additional support for decreased bone tissue adjacent to the defect.

In this treatment group, at the level of the maxillary sinus floor, a rigid heterologous cortical sheet with a thickness of 1 mm was inserted and oversized by about 2 mm in comparison to the preexisting defect [20,21]. This was performed after the placement of a heterologous cortico-cancellous graft covered with resorbable collagen membranes (Figure 3). A thermoplastic gel (TSV gel Osteobiol^®^, Turin, Italy) or pins were used to support the cortical sheet.

Following the flap’s repositioning, a continuous crestal suture and two horizontal mattress stitches were used to close the flap. After separation, a raised cut in the mesial quarter region was used to perform this suture.

Non-absorbable 4/0 threads in pseudomonofilament of polyamide were utilized for the sutures in both cases to achieve closure by primary intention.

Patients were discharged after 1 to 2 days, depending on the extension of the surgical treatment. Patients were treated with intravenous antibiotics during hospitalization (Cefazolin, 1 g twice per day) and oral antibiotics at home (amoxicillin with clavulanic acid, 1 g twice per day for 7 days). 

Patients were invited to use nasal washes with 0.9% Na solution at least thrice per day in the first post-operative month; oral anti-histaminic drugs, such as bilastine or fexofenadine hydrochloride, and nasal oils were advised to alleviate the initial post-operative nasal discomfort. All patients received detailed instructions concerning oral hygiene.

The success of the treatment was determined by the closure of the fistula at 30 days and absence of signs and symptoms of sinusitis at 30 and 90 days after the procedure.

## 3. Results

At the end of our selection process, 41 patients were consecutively enrolled. The median age was 54; (28–71 years). The male-to-female ratio was 0.8 (19 and 23, respectively). 

Considering the timing of the pathology, 9 patients presented acute sinusitis, while 33 presented the chronic form of pathology (more than 3 months). 

A total of 1 case was classified as a pre-implantological complication, 14 cases as implantological complications, and 26 cases as classical complications according to the Felisati classification. 

Two patients were treated with the fractioned combined approach, 13 patients were treated with the oral approach only, and 26 patients were treated with the combined (oral and nasal) simultaneous approach. The osteomeatal complex was obstructed in 31 cases. No major complication was registered, and no revision surgery was necessary. There was a complete resolution of the symptoms and closure of the fistula in all the patients enrolled.

## 4. Discussion

Odontogenic sinusitis deserves a distinct clinical approach compared to “classic” rhinosinusitis in light of the different pathophysiology, clinical presentation, and management. The absence of specific guidelines requires the development of standardized classification criteria. Moreover, the heterogeneity of these patients needs a multi-level classification able to be customized for the individual patient. Given the different clinical presentations and the combined clinical and surgical approaches, a multidisciplinary team should be involved when a dental origin of the sinusitis is suspected. The roles of both the oral surgeon and the otolaryngologist are extremely important during the decision process. The possibility to assess each patient from different perspectives allows us to better customize OS management, leading to an adequate success rate and avoiding surgical overtreatment.

In this study, we tried to classify all patients according to the clinical presentation, the commonly associated conditions, the specific etiology, and the treatment in order to reveal new aspects of this peculiar condition. Considering these several aspects of OS, we were able to highlight the incredible heterogeneity of these patients. Each patient is different from the others, and several features of this condition should be considered in order to provide the best OS management and the proper clinical resolution.

The distinction between acute and chronic OS highlighted some differences in terms of surgical treatment. In addition, the OAC assumes an important role in OS management, and its presence should be considered in patients’ classification. 

In the case of acute clinical presentation, the intra-oral surgical approach was found to be necessary for the treatment of the oro-antral communication (OAC). In some cases, the mucosal closure of the OAC with the removal of the dental cause of the infection was found to be adequate for a clinical resolution. 

There are many techniques for Oro-Antral fistula closure. In general, spontaneous closure of the fistula may occur if the fistula is smaller than 3 mm in diameter. If the size of the fistula is between 3 and 5 mm, scarring and suturing of the surrounding gingiva might be an efficient means. Meanwhile, surgical treatment is usually recommended if the entrance of the fistula tract is larger than 5 mm.

The buccal sliding flap introduced by Moczair is an alternative procedure for closing alveolar fistulae by shifting the flap distally by about one tooth distance. This flap technique has the advantage that the influence of buccal sulcus depth is minimal [4].

Borgonovo et al. proposed the use of the buccal flap for the closure of oroantral fistulae of moderate size, provided that they are not too posteriorly located; the palatal flap is best used in the case of fistulae located in the premolar teeth area; and the buccal flap combined with displacement of the buccal fat pad (BFP) is appropriate for fistulae located in the third molar area [22]. Additional alternatives include free connective tissue grafts (CTG) in the premolar area, free gingival grafts (FGG) from the palate, and pedicled connective tissue grafts (CTG) in the molar area. Because the depth of the vestibulum stays in its original position, these techniques should be favored considering future implantation [23]. It is also possible to use alloplastic, biological, or synthetic material for the closure of OAF. Bone grafts are recommended for the closure of chronic OAF when soft tissue flap closure fails [24]. In our study, we chose to use xenografts when a primary closure with a local flap was not sufficient, due to the high degree of predictability, easiness of use, and constant availability of the materials for reconstruction.

Considering the nasal approach, FESS was needed in 28 patients for the correction of nasal anatomical OS predisposing factors. This should not be underestimated, because FESS requires more invasive surgery, general anesthesia, and hospitalization, and, moreover, it is not exempt from complications. On the other hand, patients suffering from chronic OS with OAC underwent FESS to resolve the anatomical conditions predisposing to sinusitis, which represents a cause of the OS’s chronic persistence. If these anatomical abnormalities are not corrected with surgery, the osteomeatal complex will not resume its physiological function, leading to a greater probability of recurrence regardless of an adequate closure of the OAC. In addition, a proper nasal function may lead to a reduction in the risk of reactive sinusitis after implantological treatments [25]. In addition, if an oro-antral fistula coexisted, a combined approach was performed to remove the fistulous tract (fistulectomy) and to close the communication with a mucosal flap. The presence of an OAC is related to increased symptomatology referred to by patients, probably determined by the continuous passage of microorganisms from the oral cavity. On the other hand, our study showed that OACs do not always require surgical closure. The resolution of acute OS with spontaneous communication closure was more frequent in patients with a free osteomeatal complex and in patients in whom the communication size was relatively small (<1 cm).

In our patients for the oral approach, a mini-invasive Caldwell–Luc surgery was used, alone or together in an endoscopic nasal approach, for a complete removal of the maxillary sinus infection and odontogenic cause of sinusitis [26]. For more than a century, the Caldwell–Luc (CL) operation has been used as a surgical approach for maxillary sinus diseases. It was first described by George Caldwell in 1893 and Henri Luc in 1897 [27,28]. In our experience, we used a mini-invasive Caldwell–Luc with a mini-canine fossa antrostomy (MA). In the literature, there are numerous studies describing this technique; Muammer Melih Sabhin et al. published a retrospective study on 94 patients operated on with a combined approach of FESS and Caldwell–Luc with two different techniques: radical antrostomy (RA) and mini-canine fossa antrostomy (MA) in which it was evident that CL surgery (MA o RA) in otorhinolaryngology practice has mostly been required to provide easy access to the maxillary sinus when ESS alone would be inadequate [29]. So we can say that the literature suggests several treatment options for odontogenic sinusitis. The Caldwell–Luc approach is now only used when better access to the sinus is required, for example, for removing large foreign bodies [30].

In the literature, other approaches that were not used in our study were also described, such as the endoscopic canine fossa puncture (CFP), described by Silvia Albu et al., in which a trocar was positioned superior and lateral to the root of the upper canine, parallel to the sagittal plane to penetrate the maxillary [31]. Additionally, with this approach, excellent results were obtained, but in our experience, only the CL with mini-canine fossa antrostomy was used. After the removal of the odontogenic cause, none of the patients had complications such as failed closure of the oro-antral fistula or alterations of the mucosal flap.

The endoscopic surgical approach is different in the case of OS. In our cohort, a wide middle meatal maxillary antrostomy and the anatomical abnormalities correction were adequate for OS resolution regardless of other sinuses’ involvement. Given the prominent role of the maxillary sinus in these patients’ disease, the ethmoid or frontal sinuses surgical treatment was not necessary [32]. Although only a small amount of our patients suffered from extensive sinusitis, the possibility of removing the dental source of the infection and the restoration of adequate maxillary sinus drainage led to a resolution of contiguous sinusitis as already described in the literature. This highlights even more that OS represents a different condition compared to “classic” sinusitis, needing a customized surgical approach.

## 5. Conclusions

In our study, all 41 patients treated had surgical success with different approaches. Although some methodological limitations related to the retrospective study design and the relatively small sample exist, our study demonstrates the challenge of treating those patients suffering from OS with OACs/OAFs that are complex and heterogeneous.

A multidisciplinary approach for the correct diagnosis of odontogenic sinusitis and its involvement in the sinus in the presence of an oroantral fistula is advisable to avoid inadequate treatment and to enhance the chance of success.

In our study, it was also possible to note how the mini Caldwell–Luc approach, which for many is an outdated method since nasal endoscopy was introduced, is instead fundamental in selected cases.

In conclusion, in the endoscopic era, rather than being the main treatment modality in otorhinolaryngology practice, the CL procedure is often used to provide easy access to the maxillary sinus when ESS alone would be insufficient.

## Figures and Tables

**Figure 1 medicina-59-00937-f001:**
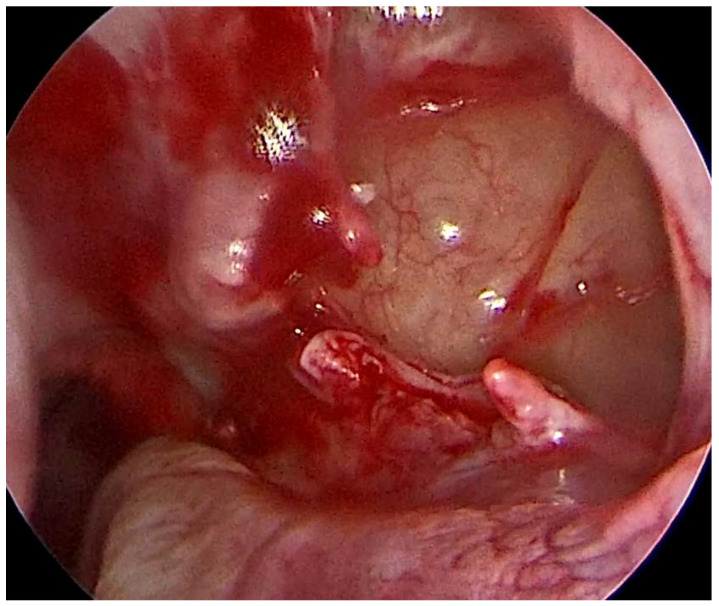
Opening of the maxillary sinus with antrostomy.

**Figure 2 medicina-59-00937-f002:**
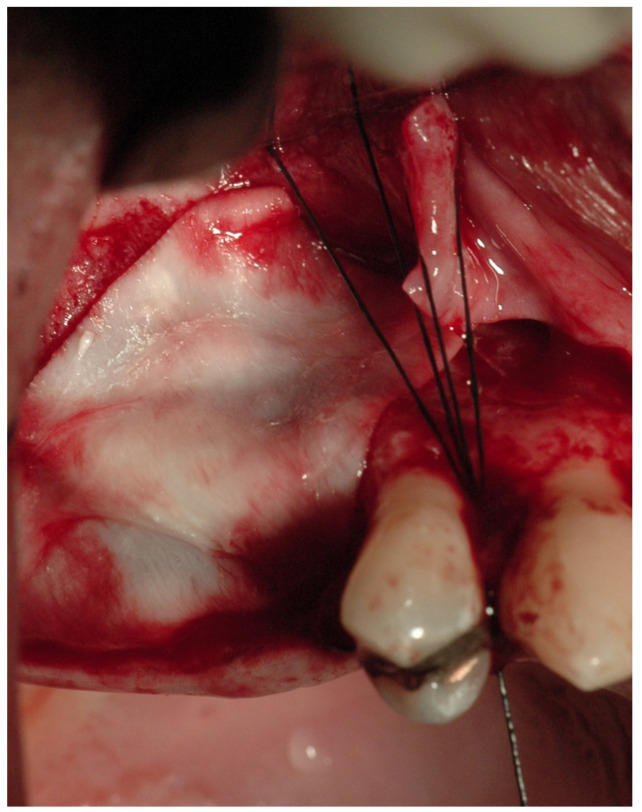
Technique with cortico-cancellous graft covered with resorbable collagen membranes.

**Figure 3 medicina-59-00937-f003:**
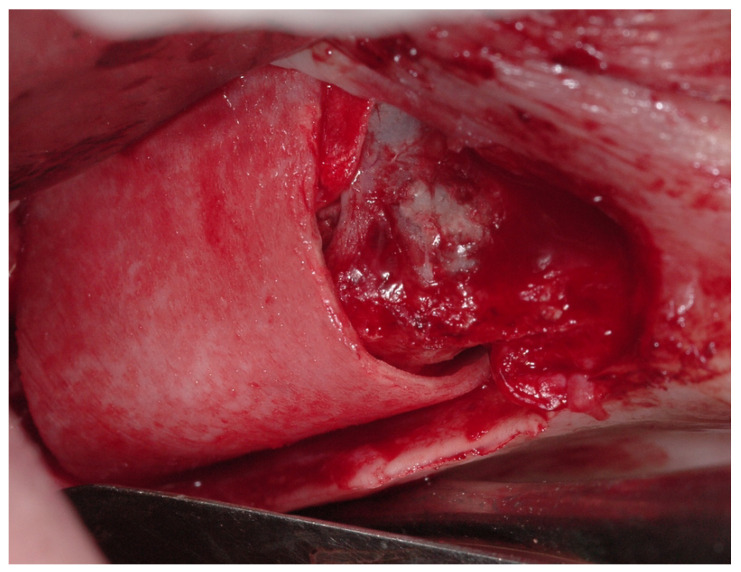
Opening of the maxillary sinus with antrostomy.

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
