# Peer review of "Odontogenic Sinusitis with Oroantral Communication and Fistula Management: Role of Regenerative Surgery"

_medicina, 2023, doi:10.3390/medicina59050937_

Round 1

Reviewer 1 Report

Thanks for interesting techniques orientated manuscript.

The fig 1 and fig 2 with chart are unnecessary. Line 305, reference is not cited in the reference list. Ref 25 does not correspond with the with line 305, 306.

Some paragraphs in discussion may be moved to introduction.  

Author Response

Dear reviewer, thank you for your evaluation of our manuscript.

We accept all the criticism raised, and the manuscript was changed accordingly.

In particular:

· English language was revised by an UK collegue.

· We cancelled fig. 1 and 2 as requested

· We fixed reference at line 305 and correspondence of references, we also added 2 references about cortical lamina and other 2 references as suggested by the other reviewer

· We moved some paragraphs about medical therapy of odontogenic sinusitis from discussion to introduction, as suggested

· We changed the title to be more suitable with the special issue

Reviewer 2 Report

The authors mad excellent job. Study design, analysis, content and methods are of high standard. I have only a minor modification to ask for. Malignant neoplasms that are complicated with a maxillary sinusitis, are a total different story. Tumor surgery for these cases has other goals and other guidelines. Quality of life after these surgery is measured with other tools. Therefore, I believe that these should be mentioned thoroughly. These two articles contain useful information and can be cited.

1) Deckard NA, Harrow BR, Barnett SL, Batra PS. Comparative analysis of quality-of-life metrics after endoscopic surgery for sinonasal neoplasms. Am J Rhinol Allergy. 2015 Mar-Apr;29(2):151-5. doi: 10.2500/ajra.2015.29.4137. PMID: 25785758.

2) Chow VJ, Tsetsos N, Poutoglidis A, Georgalas C. Quality of life in sinonasal tumors: an up-to-date review. Curr Opin Otolaryngol Head Neck Surg. 2022 Feb 1;30(1):46-57. doi: 10.1097/MOO.0000000000000774. PMID: 34889851.

Author Response

Dear reviewer, thank you for your evaluation of our manuscript.

We accept all the criticism raised, and the manuscript was changed accordingly.

In particular:

· We specified in the manuscript that the patients included were affected by non-malignant odontogenic sinusitis, and other entities were excluded ( lines 104-105)

· We cited the suggested articles, and added 2 citations about cortical lamina

· We changed the title to be more suitable with the special issue